mathematical modelling/health and disease and epidemiology/biomathematics

meta-population modelling, transmission of SARS-CoV-2, strategies for optimal control and minimal final size

**Author for correspondence:**
Zhilan Feng
e-mail: fengz@purdue.edu

# Optimal allocation of resources to healthcare workers or the general populace: a modelling study

MyVan Vo[1], Joshua A. Glasser[2] and Zhilan Feng[1,3]

[1]Department of Mathematics, Purdue University, West Lafayette, IN 47907, USA
[2]Department of Emergency Medicine, and Department of Pediatrics, Penn State School of Medicine, 500 University Drive, Hershey PA 17033, USA
[3]Division of Mathematical Sciences, National Science Foundation, Alexandria, VA, USA

JAG, 0000-0002-1778-2526; ZF, 0000-0002-6515-2510

We consider a model that distinguishes susceptible; infected, but not yet infectious; pre-symptomatic, symptomatic, asymptomatic, and hospitalized infectious; recovered and dead members of two groups: healthcare workers (HCW) and members of the community that they serve. Because of the frequency or duration of their exposures to SARS-CoV-2, a greater fraction of HCW would experience severe COVID-19 symptoms that require medical care, which reduces mortality rates, absent personal protective equipment (PPE). While N95 masks (and, possibly, other scarce medical resources) are available to members of both groups, they do not use them equally well (i.e. efficacy and compliance differ). We investigated the optimal allocation of potentially scarce medical resources between these groups to control the pandemic and reduce overall infections and mortality via derivation and analysis of expressions for the reproduction numbers and final size. We also simulated prevalence and cumulative incidence, quantities relevant to surge capacity and population immunity, respectively. We found that, under realistic conditions, the optimal allocation is virtually or entirely to HCW, but that allocation of surplus masks and other medical resources to members of the general community also reduces infections and deaths.

## 1. Introduction

At the beginning of the ongoing outbreak of COVID-19, and periodically thereafter, healthcare facilities have experienced critical shortages of personal protective equipment (e.g. N95 masks) and other medical supplies (e.g. antiseptic wipes, hand sanitizer, etc.) partly because of competition from members of

the general community. While some have a legitimate need (e.g. home healthcare), the risks to family members who are ill or those who care for them generally are much less than those to which patients or professional healthcare workers (HCW) are routinely exposed. Moreover, by virtue of their training, HCW are more knowledgeable about the risks associated with caring for patients and compliant with infection-control protocols, which ensure that they use PPE more effectively than members of the general community [1]. These observations motivated us to consider the optimal allocation of such equipment when scarce.

To address this question, we modelled the transmission of SARS-CoV-2 in a meta-population composed of two groups, HCW and others. We modified the $I$ and $R$ compartments of the familiar model in which the population is partitioned into those who are susceptible to infection, $S$; who have been infected, but are not yet infectious, $E$; are infectious, $I$; and have been removed from the infection process, $R$; to account for features of COVID-19 that affect transmission. These are the prevalence of pre- and asymptomatic infections, which make masks especially useful for preventing infection of susceptible and by infectious people, the hospitalization of some people with symptomatic infections, and disease-induced mortality. We derived expressions for the reproduction numbers and final size, which permitted us to determine the optimal allocations for reducing the effective reproduction number, total infections and deaths under scenarios differing in mask supply and proportions of the population that are HCW.

The organization of this paper is as follows: in §2, we describe the two-group model with preferential mixing. §3 outlines derivations of expressions for the basic and control reproduction numbers and final epidemic size. The allocation strategies are presented in §4, §5 collects the main results of this study, and §6 includes brief discussions of the findings. Derivations and other results are in the appendix.

## 2. The two-group SEIR epidemic model

We divided the total population into two sub-populations (or groups), with groups 1 and 2 representing the general community and HCW, respectively. Assume that each group $i$ is divided into six epidemiological classes denoted by $S_i$ (susceptible), $E_i$ (exposed), $I_{ia}$ (asymptomatic), $I_{ip}$ and $I_{is}$ (pre-symptomatic and symptomatic, respectively), $I_{ih}$ (hospitalized and isolated), $R_i$ (recovered and immune from re-infection) and $M_i$ (dead). Thus, the group population size is $N_i = S_i + E_i + I_{ia} + I_{ip} + I_{is} + I_{ih} + R_i$ ($i = 1, 2$). Because we are concerned with a single disease outbreak, births and deaths due to other causes are ignored.

One of the key differences between groups is that the contact rate is much higher among HCW than within the general community [2]. Letting $a_i$ denote the *per capita* contact rate, $a_1 < a_2$. The parameter $\beta$ denotes the probability that a susceptible person becomes infected upon contacting an infectious person. Other model parameters include the proportion $p$ of exposed people that become asymptomatic at *per capita* rate $k_a$, and proportion $1 - p$ that become symptomatic at rate $k_s$ ($1/k_j$ represents the mean latent period for the respective class $j$). For symptomatic people, $\theta_i$ and $1 - \theta_i$ represent the proportions that recover and die, respectively, and $\phi_j$ is the *per capita* rate of disease-induced mortality for class $j$. The parameters $\gamma_a$ and $\gamma_s$ are the *per capita* recovery rates of the respective classes (i.e. $1/\gamma_j$ represents the mean infectious period for people of group $j$ recovering at home) and $\delta$ is the rate of hospitalization of symptomatic people (i.e. $1/\delta$ denotes the mean interval from entering the $I_{is}$ class to hospitalization). For hospitalized people, $\theta_i$ denotes the proportion that are discharged and $1 - \theta_i$ the proportion that die, and $\phi_j$ is the death rate. The scaling constants $\eta_w$ ($w = a, p, h$) represent the infectivity of other infectious classes relative to $I_{is}$. Definitions of these parameters and the values used in numerical simulations are tabulated (table 1). A transition diagram is shown in figure 1.

The model is described by the following differential equations:

$$\left.\begin{aligned}
S_i' &= -S_i \lambda_i(t), \\
E_i' &= S_i \lambda_i(t) - \left[k_a p + k_s(1 - p)\right] E_i, \\
I_{ia}' &= k_a p E_i - \gamma_a I_{ia}, \\
I_{ip}' &= k_s(1 - p) E_i - \xi I_{ip}, \\
I_{is}' &= \xi I_{ip} - \left[\theta_i \gamma_s + \delta + (1 - \theta_i)\phi_s\right] I_{is}, \\
I_{ih}' &= \delta I_s - \left[\theta_i \gamma_h + (1 - \theta_i)\phi_h\right] I_{ih}, \\
R_i' &= \gamma_a I_{ia} + \theta_i \gamma_s I_{is} + \theta_i \gamma_h I_{ih} \\
M_i' &= (1 - \theta_i)\phi_s I_{is} + (1 - \theta_i)\phi_h I_{ih}, \quad i = 1, 2,
\end{aligned}\right\} \tag{2.1}$$

and

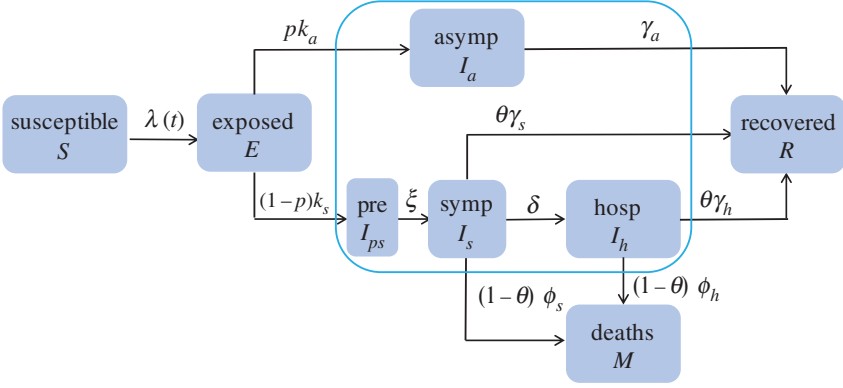

**Figure 1.** Transmission diagram for one subpopulation in a modified susceptible, exposed, infectious and recovered meta-population model whose *I* class is partitioned into pre-symptomatic $I_p$, symptomatic $I_s$, asymptomatic $I_a$ and hospitalized $I_h$ classes, and whose symptomatic and hospitalized patients may die.

**Table 1.** Definitions, notations and values of parameters of the ODE system. Notes: $i = 1, 2$ represent the general community and healthcare workers. The time unit is days. All rates are *per capita*.

| symbol | definition | values | ref. |
|---|---|---|---|
| $p$ | proportion of asymptomatic infectious ($0 \leq p \leq 1$) | 0.4 | [3] |
| $k_a$ | rate of progression from exposed $E$ to asymptomatic, $I_a$ | 1/4 | [3] |
| $k_s$ | rate of progression from exposed $E$ to pre-symptomatic, $I_p$ | 1/2 | [3] |
| $\xi$ | rate of progression from pre-symptomatic to symptomatic | 1/2 | [3] |
| $\gamma_a$ | rate of recovery of asymptomatic people | 1/7 | [3] |
| $\gamma_s$ | rate of recovery of symptomatic people | 1/7 | [3] |
| $\gamma_h$ | rate of hospital discharge | 1/14 | [3] |
| $\delta$ | rate of hospitalization of symptomatic people | 1/40 | [3] |
| $\theta_i$ | proportion of symptomatic people recovering (or dying, $1 - \theta_i$) | 0.9, 0.95 | [3] |
| $c_i$ | proportion complying with mask wearing recommendations | 0.75, 0.975 | [3] |
| $b_{iS}$ | reduced susceptibility (efficacy) by virtue of mask wearing | 0.5, 0.95 | [3] |
| $b_{iI}$ | reduced infectivity (efficacy) by virtue of mask wearing | 0.5, 0.9 | [3] |
| $\eta_w$ | scaling constants representing infectivity of $I_w$ relative to $I_s$ ($w = a, p, h$) | 0.5, 0.5, 0.1 | [3] |
| $\phi_s, \phi_h$ | rates of death among symptomatic and hospitalized people | 0.015, 0.03 | [3] |
| $a_i$ | contact rate | 10, 20 | [2,4,5] |
| $\epsilon_i$ | proportion of contacts with members of one's own group | 0.8, 0.1 | [6,7] |

where the force of infection for group $i$ is given by

$$\lambda_i(t) = \Psi_{iS} a_i \beta \sum_{j=1}^{2} c_{ij} \Psi_{jI} \frac{\eta_a I_{ja} + \eta_p I_{jp} + I_{js} + \eta_h I_{jh}}{N_j}, \quad i = 1, 2, \tag{2.2}$$

and

$$\Psi_{iS} = f(z_i, c_i, b_{iS}) \quad \text{and} \quad \Psi_{iI} = g(z_i, c_i, b_{iI}) \tag{2.3}$$

represent reductions in susceptibility and infectivity, respectively, due to mask wearing, where $z_i$ is the proportion of masks available to members of group $i$, $c_i$ is the proportion complying (compliance) with mask-wearing, and $b_{iS}$ and $b_{iI}$ denote reduced susceptibility and infectivity (efficacy) by virtue of mask-wearing. Symbols not defined in table 1 are included in table 2 in the main text or table 4 in appendix E. These functions can be thought of as weighted average

susceptibility and infectiousness among those with and without masks

$$\left.\begin{array}{l} \Psi_{iS} = z_i(1 - c_i b_{iS}) + (1 - z_i) = 1 - z_i c_i b_{iS} \\ \Psi_{iI} = z_i(1 - c_i b_{iI}) + (1 - z_i) = 1 - z_i c_i b_{iI}. \end{array}\right\} \tag{2.4}$$

and

The initial conditions are

$$S_i(0) = S_{i0}, \quad E_i(0) = E_{i0}, \quad R_i(0) = M_i(0) = 0, \quad I_{iw}(0) = I_{iw0}, \quad w = s, a, p, h, \quad i = 1, 2. \tag{2.5}$$

In Model (2.1), contacts between sub-groups are described by mixing matrix $C = (c_{ij})$, where $c_{ij}$ is the proportion of the $i$th sub-group's contacts that is with members of the $j$th and $I_j/N_j$ is the probability that a randomly encountered member of group $j$ is infectious. The elements of $C$ have the following form:

$$c_{ij} = \epsilon_i \delta_{ij} + (1 - \epsilon_i) f_j, \quad \text{where } f_j = \frac{(1 - \epsilon_j) a_j N_j}{\sum_k (1 - \epsilon_k) a_k N_k}, \tag{2.6}$$

where the $\epsilon_i \in [0, 1]$ describe preference for one's own group and $\delta_{ij}$ is the Kronecker delta (i.e. 1 when $i = j$ and 0 otherwise). Letting $q$ represent the 0.03–0.05 of the total population that is healthcare workers (Bureau of Labor Statistics [6,7]), the number of people in the general community, $N_1$, is $N(1 - q)$ and the number of HCW, $N_2$, is $Nq$. Thus, in the case of proportionate mixing,

$$c_{11} = c_{21} = \frac{a_1(1 - q)}{a_1(1 - q) + a_2 q}$$

and

$$c_{12} = c_{22} = \frac{a_2 q}{a_1(1 - q) + a_2 q}.$$

All parameters are non-negative.

# 3. Reproduction numbers and final size relation

## 3.1. Reproduction numbers

In this section, we provide biological interpretations of the elements of the next-generation matrix (NGM). Derivations can be found in appendix A.

For ease of presentation, we introduce additional notations for biologically meaningful quantities, defined in table 2. In appendix A, we show that the NGM can be written as

$$K_{11} = \begin{pmatrix} a_1 \beta A_{11} & a_1 \beta A_{12} \\ a_2 \beta A_{21} & a_2 \beta A_{22} \end{pmatrix},$$

where

$$A_{ij} = \pi_{ij} c_{ij} \left( \eta_a p^A \tau_j^A + \eta_p p^P \tau^P + p^P \tau_j^S + \eta_h p^P p_j^H \tau_j^H \right)$$

represents the effective contacts with people in group $i$ during the infectious period of a person who was infected while in group $j$. Then $a_i \beta A_{ij}$ represents the average number of new infections generated among susceptible people in group $i$ by a person infected in group $j$.

Let $A = a_1 \beta A_{11}$, $B = a_1 \beta A_{12}$, $C = a_2 \beta A_{21}$ and $D = a_2 \beta A_{22}$. Then $\mathcal{R}_C$ is given by the dominant eigenvalue of $K_{11}$,

$$\mathcal{R}_C = \frac{1}{2} \left[ A + D + \sqrt{(A - D)^2 + 4BC} \right]. \tag{3.1}$$

In the case of proportionate mixing, $\mathcal{R}_C = A + D$.

Considering $\mathcal{R}_C(\mathbf{z})$ as a function of $\mathbf{z} = (z_1, z_2)$, the basic reproduction number $\mathcal{R}_0$ is given by $\mathcal{R}_C(0)$ (i.e. in the absence of mask use), in which case $\Psi_{iS} = \Psi_{iI} = 1$. Notice that the basic reproduction numbers for the two sub-populations in isolation are

$$\left.\begin{array}{l} \mathcal{R}_{01} = a_1 \beta (\eta_a p^A \tau^A + \eta_p p^P \tau^P + p^P \tau_1^S + \eta_h p^P p_1^H \tau_1^H) \\ \mathcal{R}_{02} = a_2 \beta (\eta_a p^A \tau^A + \eta_p p^P \tau^P + p^P \tau_2^S + \eta_h p^P p_2^H \tau_2^H). \end{array}\right\} \tag{3.2}$$

and

**Table 2.** Definition of symbols used in the expressions for $\mathcal{R}_C$ and $\mathcal{R}_{0i}$. Note: $i = 1,\ 2$ represent the general community and HCW, respectively.

| symbol | description | definition |
|---|---|---|
| $\tau^E$ | latent period | $(k_a\, p + k_s\,(1-p))^{-1}$ |
| $\tau^A$ | asymptomatic infectious period | $\gamma_a^{-1}$ |
| $\tau^P$ | pre-symptomatic infectious period | $\xi^{-1}$ |
| $\tau_i^S$ | hospitalized infectious period while in group $i$ | $(\theta_i\gamma_s + \delta + (1-\theta_i)\phi_s)^{-1}$ |
| $\tau_i^H$ | symptomatic infectious period while in group $i$ | $(\theta_i\gamma_h + (1-\theta_i)\phi_h)^{-1}$ |
| $p^A$ | probability of being asymptomatic | $p k_a \tau^E$ |
| $p^P$ | probability of being pre-symptomatic | $(1-p)k_s\tau^E$ |
| $p_i^H$ | probability of being hospitalized while in group $i$ | $\delta\tau_i^S$ |
| $\Psi_{iS}$ | reductions in susceptibility due to mask wearing in group $i$ | $1 - z_i\, c_i\, b_{iS}$ |
| $\Psi_{il}$ | reductions in infectivity due to mask wearing in group $i$ | $1 - z_i\, c_i\, b_{il}$ |
| $\pi_{ij}$ | reduction in FOI of group $i$ due to mask wearing of group $j$ | $\Psi_{iS}\Psi_{jl}$ |

## 3.2. The final size relation

To simplify this presentation, we assume that

$$S_{i0} = N_i - E_{is0}, \quad E_{is0} > 0, \quad I_{iw0} = R_i(0) = M_i(0) = 0, \quad w = s, a, p, h, \quad i = 1, 2. \tag{3.3}$$

Let $Z_i = S_i(0) - S_i(\infty)$. Then, as we show in appendix B,

$$Z_i = S_i(0)\left(1 - \exp\left[-\sum_{j=1}^{2} a_i\beta A_{ij}\frac{Z_j + E_{j0}}{N_j}\right]\right), \quad i = 1, 2.$$

Letting $N = N_1 + N_2$ denote the total population size and $\mathcal{F}_i = Z_i/N$ the fraction infected in group $i$, the final epidemic size $\mathcal{F}$ is given by

$$\mathcal{F} = \mathcal{F}_1 + \mathcal{F}_2 = \frac{Z_1}{N} + \frac{Z_2}{N}.$$

For the case when $E_{i0} = I_{iw0} = 0$ ($w = a, p, h$) and $I_{is0} > 0$, the final size relation for group $i$ is

$$Z_i = S_i(0)\left(1 - \exp\left[-\sum_{j=1}^{2} a_i\beta\frac{A_{ij}Z_j + c_{ij}\pi_{ij}\tau_j^S I_{js0} + c_{ij}\pi_{ij}\eta_h\tau_j^H p_j^H I_{js0}}{N_j}\right]\right), \quad i = 1, 2. \tag{3.4}$$

## 3.3. The total number of deaths

Consider the initial conditions (3.3) and the $M_i$ equation in system (2.1). It follows that

$$M_i(t) = (1-\theta_i)\phi_s\int_0^t I_{is}(u)\mathrm{d}u + (1-\theta_i)\phi_h\int_0^t I_{ih}(u)\mathrm{d}u. \tag{3.5}$$

Let $\mathcal{M}_i = M_i(\infty)$ denote the total number of deaths in group $i$. Then, from (3.5) and (B 2), we have

$$\mathcal{M}_i = (Z_i + E_{i0})(1-\theta_i)[\phi_s\tau_i^S p^P + \phi_h\tau^H p^P p_i^H]. \tag{3.6}$$

# 4. Allocation strategies

Assume that $\mathcal{R}_C > 1$ and $mN$ supplemental masks are available, where $m$ is the proportion available to the total population. Let $(z_1, z_2)$ be vectors of the proportions for whom masks are available in the general community and HCW. We compare the following three allocation strategies in terms of their reductions in $\mathcal{R}_C$:

**A**: $(z_1, z_2) = (m, m)$ (*Proportional*). Both groups receive masks in proportion to their group sizes.

**B**: $(z_1, z_2) = (\tilde{z}_1, 1)$ (*Full Coverage*). All HCW receive masks with any remaining being given to the general community, $\tilde{z}_1 = ((mN - N_2)/N_1) = (m - q)/(1 - q)$.

**C**: $(z_1, z_2) = (\hat{z}_1, \hat{z}_2)$ (*Optimal*). This is the case where $\mathcal{R}_C(\hat{z}_1, \hat{z}_2) = \min \mathcal{R}_C(z_1, z_2)$.

The effects of Strategies **A** and **B** on reducing $\mathcal{R}_C$ can be computed numerically. Strategy **C** $(\hat{z}_1, \hat{z}_2)$ can be obtained using the gradient approach [8], which is the solution to the following Lagrange optimization problem:

$$\text{Minimize } \mathcal{R}_C(z_1, z_2) \quad \text{subject to } z_1 N_1 + z_2 N_2 = mN.$$

We can get $\hat{z}_1$ and $\hat{z}_2$ by solving simultaneously the equations $\nabla \mathcal{R}_C + \lambda(N_1, N_2) = 0$ and $z_1 N_1 + z_2 N_2 = mN$, where $\nabla \mathcal{R}_C$ is the partial derivative of $\mathcal{R}_C$ with respect to $(z_1, z_2)$ and $\lambda$ is a Lagrange multiplier.

Let $\mathcal{R}_{Cmin} = \mathcal{R}_C(\hat{z}_1, \hat{z}_2)$. To facilitate comparison, we consider the following quantities to evaluate the *Optimal* Strategy **C** relative to the others in terms of either $\mathcal{R}_C$:

$$\Delta^{\mathcal{R}}_{(z_1, z_2)} = \frac{\mathcal{R}_C(z_1, z_2) - \mathcal{R}_{Cmin}}{\mathcal{R}_C(z_1, z_2)} \tag{4.1}$$

or final size $\mathcal{F}$:

$$\Delta^{\mathcal{F}}_{(z_1, z_2)} = \frac{\mathcal{F}(z_1, z_2) - \mathcal{F}(\hat{z}_1, \hat{z}_2)}{\mathcal{F}(z_1, z_2)}, \tag{4.2}$$

where $(z_1, z_2)$ is the allocation corresponding to a particular strategy. We will refer to $\Delta^{\mathcal{R}}_{(z_1, z_2)}$ and $\Delta^{\mathcal{F}}_{(z_1, z_2)}$ as efficacies with respect to reducing $\mathcal{R}_C$ and $\mathcal{F}$, respectively.

# 5. Results

In this section, we present numerical results illustrating various model outcomes under Strategies **A**–**C**. Parameter values used for simulations are listed in tables 1 and 2. The range for $\beta$, (0.0265, 0.0442), was determined from equation (3.1) with $z_1 = z_2 = 0$ and $\mathcal{R}_0$ from 2 to 3 [3]. The total population size is $N = 10^7$. The proportion of HCW depends on location (Bureau of Labor Statistics [6,7]); thus, we explore 1%, 3% and 5% HCW in the population, $q = 0.01, 0.03, 0.05$, respectively. Our objective is to understand the impact of allocation strategies under varying availability of supplemental masks.

## 5.1. Effect of allocation strategies on $\mathcal{R}_C$ and final size $\mathcal{F}$

Figure 2 shows the reduction in $\mathcal{R}_C(z_1, z_2)$ attained by the *Full Coverage* of HCW (Strategy **B**) and *Optimal* allocation (Strategy **C**) for nine scenarios differing in availability of supplemental masks and proportions of HCW in the population. The white star and black dot indicate Strategies **B** and **C**, respectively. We fixed $\beta$ to be 0.0332, which—when 5% of the population are HCW—corresponds to $\mathcal{R}_0 = 2.5$; however, if we varied $\beta$, the *Optimal* allocation would not change. Figure 2 illustrates that, as mask availability increases, the *Optimal* Strategy **C** moves closer to the *Full Coverage* Strategy **B**. When masks are abundant, strategies **B** and **C** reduce $\mathcal{R}_C$ similarly. However, when masks are scarce, the *Optimal* allocation is necessary to minimize $\mathcal{R}_C$.

Figure 3 shows the efficacy of the *Optimal* Strategy **C** with respect to the *Proportional* Strategy **A** and *Full Coverage* Strategy **B** using the quantities $\Delta^{\mathcal{R}}_{(z_1, z_2)}$ and $\Delta^{\mathcal{F}}_{(z_1, z_2)}$ given in (4.1) and (4.2). Figure 3*a* shows that Strategy **C** is much more effective at reducing $\mathcal{R}_C$ than Strategy **A** and marginally more effective than Strategy **B**. For example, given 5% HCW and 10% supplemental masks, Strategy **C** is 11.6% more effective at lowering $\mathcal{R}_C$ than Strategy **A**, but only 0.2% more effective than Strategy **B**. The differences between strategies decrease as the availability of masks increases.

Results shown in figure 3 indicate that the strategy minimizing the control reproduction number $\mathcal{R}_C$ does not necessarily minimize the final size $\mathcal{F}$. We observe in figure 3*b* that the *Optimal* Strategy **C** is more effective at reducing final size than the *Proportional* Strategy **A**, but less effective than *Full Coverage* Strategy **B**. If 5% of the population are HCW and supplemental masks suffice for 10% of the population, for example, the *Optimal* allocation is 5.7% more effective at lowering the final size than the *Proportional* allocation and 1.5% less effective, as noted by negative efficacy, than the *Full Coverage* allocation. In other words, the *Full Coverage* allocation yields a smaller final size than the *Optimal* allocation.

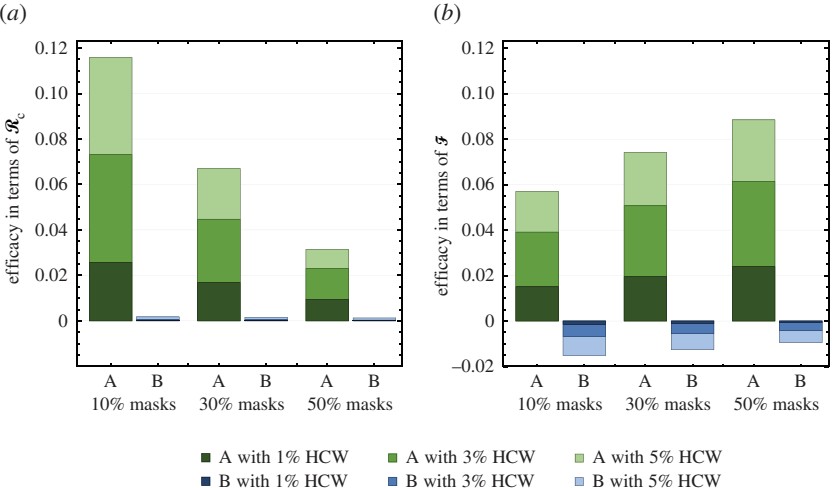

**Figure 2.** Contour plots of $\mathcal{R}_C$ with the *Optimal* allocation, $(\hat{z}_1, \hat{z}_2)$, and *Full Coverage* for HCW, $(\tilde{z}_1, 1)$. The columns are for 1%, 3% and 5% of HCW in the population and rows differ in availability of supplemental masks ($m = 0.1, 0.3, 0.5$). The intersection where the black constraint line is tangent to the red curve corresponds to the *Optimal* Strategy **C**, which is marked by a black dot. The white star corresponds to the *Full Coverage* Strategy **B**. The top-right plot (10% supp masks and 5% HCW) shows that the optimal solution distributes 85% of the available masks to HCW and the remainder to community, enough for approximately 6% of that population.

**Figure 3.** Comparison of allocation strategies. The *Optimal* Strategy **C** is compared with Strategies **A** and **B** in terms of relative reductions (efficacies) in ($a$) $\mathcal{R}_C$ and ($b$) $\mathcal{F}$ for different HCW and mask supplies. Efficacies are defined in (4.1) and (4.2).

Figure 4 shows that the *Full Coverage* Strategy **B** is best for reducing final size $\mathcal{F}$. The large red asterisk at the top-right corner indicates the case of no supplemental masks. On each curve showing a set of possible outcomes for a fixed quantity of supplemental masks, the outcomes corresponding to

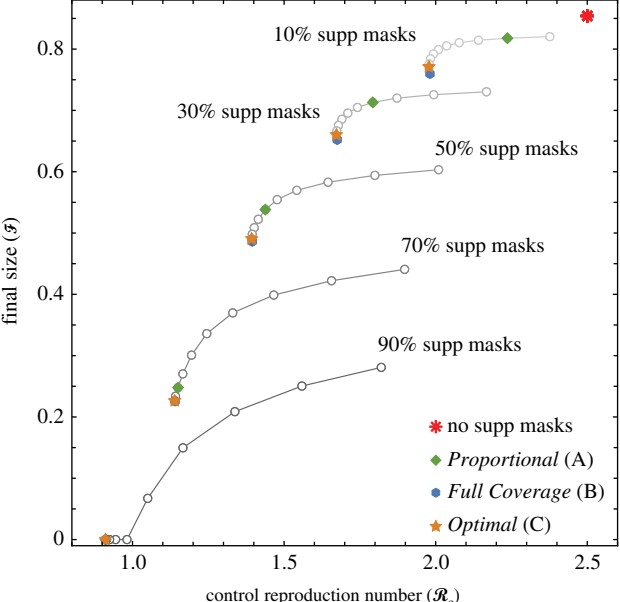

**Figure 4.** Effect of mask supply on the relationship between final size and control reproduction number. We assume that 5% of the population is HCW and that $\beta = 0.0332$ and vary the percentage of supplemental masks available. The connected sets of points correspond to outcomes given a fixed quantity of supplemental masks. The dot at the upper right of each connected set shows the allocation where no masks are given to HCW. Each successive dot represents an increase of 10% coverage for HCW until *Full Coverage* is attained.

Strategies **A–C** are labelled with different symbols. The open circles show intermediate allocations that are not considered in the analysis. Recall that figure 3 shows that, as the availability of masks increases, the differences between Strategies **B** and **C** at reducing $\mathcal{F}$ decrease.

**Remark.** For ease of presentation, figure 4 includes only five quantities of supplemental masks, for which none of the curves intersect, indicating that more masks lead to smaller final sizes and lower $\mathcal{R}_C$. However, this is not always the case. A different scenario is presented in figure 8 (in the appendix), in which 5% and 10% of supplemental masks are compared. The two curves intersect, showing that, under those allocations, the resulting $\mathcal{R}_C$ and final size $\mathcal{F}$ are the same.

Prevalence and cumulative incidence over time under the *Optimal* Strategy **C** are plotted in figure 5 for different availabilities of supplemental masks. Figure 5a,b illustrates that the first 10% of masks provide the most significant reduction in prevalence and cumulative incidence, respectively.

**Remark.** While we are interested in identifying the best strategies for mitigating the outbreak, this depends on the policy goal. Figure 3 illustrates that Strategy **C** reduces $\mathcal{R}_C$ the most, while Strategy **B** reduces $\mathcal{F}$ the most. This also implies the greatest reduction in disease-induced mortality, as deaths are nearly proportional to final size (see §3.3, equation (3.6) for the relationship between the final size and number of deaths).

## 5.2. Effects of compliance in the community

The role of compliance with mask wearing is illustrated in figures 6 and 7 for the *Optimal* Strategy **C**. We observe in figure 6 that, when compliance in the general community increases, the *Optimal* allocation of masks to the general community increases.

Figure 7 illustrates that compliance and $\mathcal{R}_C$ vary inversely regardless of allocation strategy. Thus, $\mathcal{R}_C$ decreases as compliance increases.

**Remark.** In our analyses, we assumed that HCW are 5% of the population or less. Were HCW 20% of the population, the *Optimal* allocation would still be the best strategy for reducing $\mathcal{R}_C$. However, the strategy that reduces $\mathcal{F}$ the most is no longer the *Full Coverage* Strategy **B**, but rather is between Strategies **A** and **C**. The best strategy still allocates more masks to HCW than members of the general community.

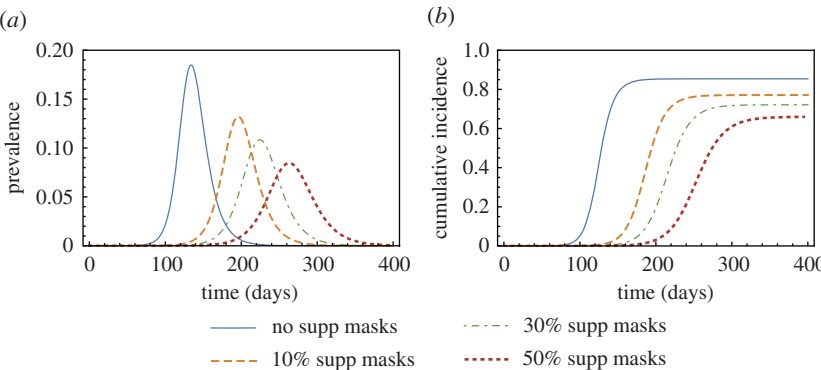

**Figure 5.** Prevalence and cumulative incidence under *Optimal* Strategy **C**. The plots in (*a*) and (*b*) show the effect of mask supply on the prevalence and cumulative incidence, respectively, for different availabilities of supplemental masks ($m = 0, 0.1, 0.3, 0.5$).

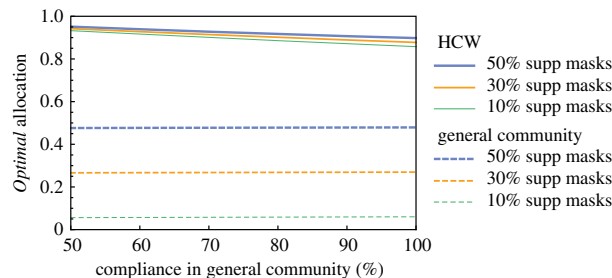

**Figure 6.** *Optimal* allocation versus compliance for different quantities of supplemental masks. Results for 5% of the population HCW ($q = 0.05$) and enough supplemental masks for 50% of the total population ($m = 0.5$). The dashed and solid lines are plots of $\hat{z}_1$ and $\hat{z}_2$ representing the *Optimal* allocations of masks under Strategy **C**.

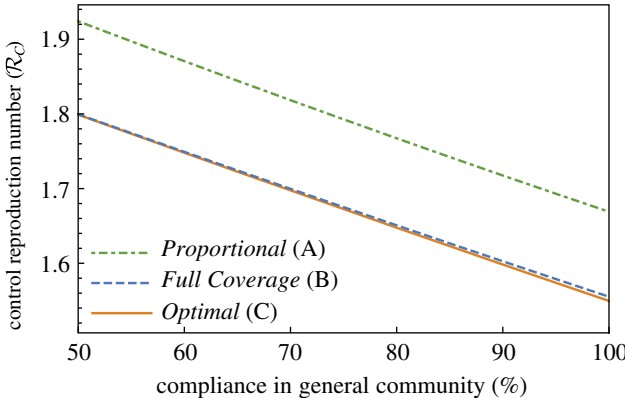

**Figure 7.** Control reproduction number as a function of compliance in community. Plot of $\mathcal{R}_C$ as a function of compliance in the general community for 30% supplemental masks ($m = 0.3$).

We assumed that the contact rate of HCW, $a_2$, is twice that of members of the general community, $a_1$. Were the ratio $a_2/a_1$ to increase, the best strategy would shift towards the *Full Coverage* allocation.

## 5.3. Sensitivity analysis

In this section, we describe sensitivity analyses of the basic, $\mathcal{R}_0$ and control reproduction number, $\mathcal{R}_C$. The analyses are based on the method of Latin hypercube sampling (LHS) with assumptions about the ranges and distributions of parameters listed in table 3. More details are included in appendix D and illustrated in figure 9. We omitted most parameters that have a minimal effect on the reproduction numbers from figure 9. Although $\theta_2$ and $c_2$ also have minimal influence, we retained

**Table 3.** Assumptions made in the sensitivity analysis. See table 1 for parameter definitions.

| symbol | mean | distribution | range |
|---|---|---|---|
| $p$ | 0.4 | uniform | (0.25, 0.55) |
| $k_a$ | 1/4 | triangular | (1/5, 1/3) |
| $k_s$ | 1/2 | triangular | (1/4, 1/3) |
| $\xi$ | 1/2 | triangular | (1/4, 1/2) |
| $\gamma_a$ | 1/7 | triangular | (1/9, 1/5) |
| $\gamma_s$ | 1/7 | triangular | (1/9, 1/5) |
| $\gamma_h$ | 1/14 | triangular | (1/18, 1/10) |
| $\theta_1, \theta_2$ | 0.9, 0.95 | triangular | (0.8, 1), (0.9, 1) |
| $\delta$ | 0.025 | triangular | (0.02, 0.03) |
| $c_1, c_2$ | 0.75, 0.975 | uniform | (0.5, 1), (0.95, 1) |
| $b_{1S}, b_{2S}$ | 0.5, 0.95 | triangular | (0.4, 0.6), (0.9, 1) |
| $b_{1I}, b_{2I}$ | 0.5, 0.9 | triangular | (0.4, 0.6), (0.85, 0.95) |
| $\eta_a$ | 0.5 | triangular | (0.3, 0.7) |
| $\eta_p$ | 0.5 | triangular | (0.3, 0.7) |
| $\eta_h$ | 0.1 | triangular | (0.05, 0.2) |
| $\phi_s$ | 0.015 | triangular | (0.01, 0.02) |
| $\phi_h$ | 0.03 | triangular | (0.02, 0.04) |

them because $\theta_1$ and $c_1$ have non-negligible effects. For the analysis of $\mathcal{R}_C$, $z_1$ and $z_2$ were fixed at (0.25, 1), although the partial rank correlation coefficients (PRCC) results show a similar pattern for other non-zero $(z_1, z_2)$ values. These results show that both $\mathcal{R}_0$ and $\mathcal{R}_C$ are most sensitive to $\beta$ and $\gamma_h$, and that the basic reproduction number is within the range of 2–3, consistent with the literature.

# 6. Discussion

As supplies of N95 masks and other items of PPE were limited globally at the beginning and locally throughout the pandemic of COVID-19, we considered their allocation to HCW and members of the general community. For this purpose, we modified a model in which the host population is partitioned into susceptible; infected, but not yet infectious; infectious; or recovered and immune to include asymptomatic, pre-symptomatic, hospitalized and dead compartments.

Our model population also comprises HCW, who typically compose no more than 5% of their populations, and others. By virtue of their vocations, HCW have a greater risk of exposure to infectious people. However, they work in medical facilities with infection-control protocols that are more stringent than public-health recommendations for the general community. And, by virtue of their training, HCW both comply more closely with those protocols and use PPE more effectively than untrained members of the general community.

Our objectives were to determine allocations of N95 masks, a proxy for PPE, that control the epidemic most expeditiously or limit the total number of infections and, all else equal, deaths. We determined these *Optimal* allocations by deriving expressions for the reproduction numbers ($\mathcal{R}_0$ and $\mathcal{R}_C$) and final size ($\mathcal{F}$), and by calculating the gradient of the effective reproduction number with respect to different allocations of limited supplies of PPE. Two other allocations are also considered (Strategies **A** and **B**) and compared with the *Optimal* allocation (Strategy **C**).

As expected, the *Optimal* allocation is best for controlling the pandemic. But we found that allocation exclusively to HCW is the most effective for reducing final size, and all else equal, deaths. That the best strategy depends on the objective almost certainly is a general result. As supplies become less limited, strategies converge, another result that almost certainly is general. Limitations include under-estimating the greater risk of exposure that HCW experience by virtue of their vocations. We simply assumed that their contact rates were twice that of members of the general community.

Future work might include cross-classification by age, a proxy for chronic conditions that increase the need for hospital-based care and mortality. Thus, elderly members of the general community or family members who care for them may warrant a greater share of PPE. Also, insofar as healthcare facilities cannot function at full capacity without a full complement of healthy HCW, and fully functioning medical facilities reduce mortality, HCW themselves could become a scarce resource. We compared various strategies via the proportions of our two populations that were infected. Owing to the smaller number of HCWs than members of the general community, however, this may not have adequately represented the impact of various strategies on that group. To consider HCWs a scarce resource, one would need to make mortality among those needing hospital-based care a function of the ability of hospitals to care for them, in turn a function of the proportion of HCWs able to work. However worthwhile these refinements may be, we do not expect them to alter our conclusions fundamentally.

Data accessibility. This article has no additional data.
Authors' contributions. M.V. performed analyses and simulations and wrote the manuscript. J.A.G. provided information and reviewed the manuscript. Z.F. contributed to study design, supervised the work, provided information and reviewed the manuscript.
Competing interests. We declare we have no competing interests.
Funding. This work is partially supported by the IR/D program from the National Science Foundation (NSF) and the NSF grant no. DMS-1814545.
Acknowledgements. This work was done in collaboration with John W. Glasser. We thank the anonymous reviewers for constructive critiques of earlier drafts of the manuscript.
Disclaimer. The findings and conclusions in this report are those of the authors and do not necessarily represent the official position of the National Science Foundation.

# Appendix A. Derivation of the reproduction numbers

For the derivation of the basic and effective reproduction numbers, we introduce the fractions $s_i(t) = S_i(t)/N_i$, $e_i(t) = E_i(t)/N_i$, $i_{iw}(t) = I_{iw}(t)/N_i$ (for $w = a$, $p$, $s$, $h$), $r_i(t) = R_i(t)/N_i$, $m_i(t) = M_i(t)/N_i$. Then, Model (2.1) becomes

$$
\left.
\begin{aligned}
s_i' &= -s_i \lambda_i(t), \\
e_i' &= s_i \lambda_i(t) - [k_a p + k_s(1-p)]e_i, \\
i_{ia}' &= k_a p e_i - \gamma_a i_{ia}, \\
i_{ip}' &= k_s(1-p)e_i - \xi i_{ip}, \\
i_{is}' &= \xi i_{ip} - [\theta_i \gamma_s + \delta_i + (1-\theta_i)\phi_s]i_{is}, \\
i_{ih}' &= \delta_i i_{is} - [\theta_i \gamma_h + (1-\theta_i)\phi_h]i_{ih}, \\
r_i' &= \gamma_a i_{ia} + \theta_i \gamma_s i_{is} + \theta_i \gamma_h i_{ih}, \\
m_i' &= (1-\theta_i)\phi_s i_{is} + (1-\theta_i)\phi_h i_{ih}, \quad i = 1, 2,
\end{aligned}
\right\}
\tag{A1}
$$

and

where the force of infection for group $i$ is given by

$$
\lambda_i(t) = \Psi_{iS} a_i \beta \sum_{j=1}^{2} c_{ij} \Psi_{jI}(\eta_a i_{ja} + \eta_p i_{jp} + i_{js} + \eta_h i_{jh}), \quad i = 1, 2.
\tag{A2}
$$

Let $s_i^* = S_i^*/N_i$ denote the fraction of susceptibles in group $i$ at the disease-free equilibrium (DFE). For Model A1, $(s_1^*, s_2^*) = (1, 1)$. Considering only the disease variables, $e_i$, $i_{ia}$, $i_{ip}$, $i_{is}$, $i_{ih}$, the NGM [9,10] is

$$
J = \begin{pmatrix} J_{11} & J_{12} \\ J_{21} & J_{22} \end{pmatrix},
$$

where

$$J_{11} = \begin{pmatrix} -(k_a p + k_s(1-p)) & 0 \\ 0 & -(k_a p + k_s(1-p)) \end{pmatrix},$$

$$J_{12} = \begin{pmatrix} a_1 \beta_1 & 0 \\ 0 & a_2 \beta_2 \end{pmatrix} \begin{pmatrix} \pi_{11} & \pi_{12} \\ \pi_{21} & \pi_{22} \end{pmatrix} \begin{pmatrix} c_{11} & c_{12} \\ c_{21} & c_{22} \end{pmatrix} \begin{pmatrix} \eta_{1a} & 0 & \eta_{1p} & 0 & 1 & 0 & \eta_{1h} & 0 \\ 0 & \eta_{2a} & 0 & \eta_{2p} & 0 & 1 & 0 & \eta_{2h} \end{pmatrix},$$

and

$$J_{21} = \begin{pmatrix} k_a p & 0 \\ 0 & k_a p \\ k_s(1-p) & 0 \\ 0 & k_s(1-p) \\ 0 & 0 \\ 0 & 0 \\ 0 & 0 \\ 0 & 0 \end{pmatrix},$$

and

$$J_{22} = \begin{pmatrix} -\frac{1}{\tau_1^A} & 0 & 0 & 0 & 0 & 0 & 0 & 0 \\ 0 & -\frac{1}{\tau_2^A} & 0 & 0 & 0 & 0 & 0 & 0 \\ 0 & 0 & -\frac{1}{\tau^P} & 0 & 0 & 0 & 0 & 0 \\ 0 & 0 & 0 & -\frac{1}{\tau^P} & 0 & 0 & 0 & 0 \\ 0 & 0 & \frac{1}{\tau^P} & 0 & -\frac{1}{\tau_1^S} & 0 & 0 & 0 \\ 0 & 0 & 0 & \frac{1}{\tau^P} & 0 & -\frac{1}{\tau_2^S} & 0 & 0 \\ 0 & 0 & 0 & 0 & \delta_1 & 0 & -\frac{1}{\tau_1^H} & 0 \\ 0 & 0 & 0 & 0 & 0 & \delta_2 & 0 & -\frac{1}{\tau_2^H} \end{pmatrix}.$$

Let $J = F - V$, where $F = \begin{pmatrix} 0 & J_{12} \\ 0 & 0 \end{pmatrix}$, $V = \begin{pmatrix} -J_{11} & 0 \\ -J_{21} & -J_{22} \end{pmatrix}$.

Note that $V^{-1} = \begin{pmatrix} -J_{11}^{-1} & 0 \\ J_{22}^{-1} J_{21} J_{11}^{-1} & -J_{22}^{-1} \end{pmatrix}$, where $-J_{11}^{-1} = \begin{pmatrix} \tau^E & 0 \\ 0 & \tau^E \end{pmatrix}$,

$$-J_{22}^{-1} = \begin{pmatrix} \tau_1^A & 0 & 0 & 0 & 0 & 0 & 0 & 0 \\ 0 & \tau_2^A & 0 & 0 & 0 & 0 & 0 & 0 \\ 0 & 0 & \tau^P & 0 & 0 & 0 & 0 & 0 \\ 0 & 0 & 0 & \tau^P & 0 & 0 & 0 & 0 \\ 0 & 0 & \tau_1^S & 0 & \tau_1^S & 0 & 0 & 0 \\ 0 & 0 & 0 & \tau_2^S & 0 & \tau_2^S & 0 & 0 \\ 0 & 0 & p_1^H \tau_1^H & 0 & p_1^H \tau_1^H & 0 & \tau_1^H & 0 \\ 0 & 0 & 0 & p_2^H \tau_2^H & 0 & p_2^H \tau_2^H & 0 & \tau_2^H \end{pmatrix},$$

and

$$J_{22}^{-1} J_{21} J_{11}^{-1} = \begin{pmatrix} p^A \tau_1^A & 0 \\ 0 & p^A \tau_2^A \\ p^P \tau^P & 0 \\ 0 & p^P \tau^P \\ p^P \tau_1^S & 0 \\ 0 & p^P \tau_2^S \\ p^P p_1^H \tau_1^H & 0 \\ 0 & p^P p_2^H \tau_2^H \end{pmatrix}.$$

The NGM is

$$K = FV^{-1} = \begin{pmatrix} J_{12} J_{22}^{-1} J_{21} J_{11}^{-1} & -J_{12} J_{22}^{-1} \\ 0 & 0 \end{pmatrix}.$$

If $K_{11} = J_{12} J_{22}^{-1} J_{21} J_{11}^{-1}$, the eigenvalues of $K_{11}$ are also the eigenvalues of $K$.

# Appendix B. Derivation of the final size relation

Note that

$$(S_i(t) + E_i(t))' = -(k_a p + k_s(1 - p))E_i(t),$$

for $i = 1, 2$. Because $S_i(t)$ and $E_i(t)$ are non-negative for all $t$ and $S_i(t) + E_i(t)$ is a non-negative decreasing function, it is bounded below and achieves a limit as $t$ approaches infinity. Given that $\int_{t=0}^{\infty} E_i \mathrm{d}t \le N_1$ and hence finite, it follows that $E_i(t)$ approaches 0 as $t$ approaches $\infty$. So, $\lim_{t \to \infty} S_i(t) = S_i(\infty)$ exists. By replacing $E_i(\infty) = 0$, we get

$$S_i(0) + E_i(0) - S_i(\infty) = (k_a p + k_s(1 - p)) \int_0^{\infty} E_i(u) \mathrm{d}u,$$

$$\int_0^{\infty} E_i(u) \mathrm{d}u = \tau^E [S_i(0) + E_i(0) - S_i(\infty)]$$

and

$$\int_0^{\infty} E_i(u) \mathrm{d}u = \tau^E [S_i(0) + E_{i0} - S_i(\infty)].$$

Using the $S_i'$ equation, we get

$$\frac{S_i'}{S_i} = -\Psi_{iS} a_i \beta \sum_{j=1}^{2} c_{ij} \Psi_{jI} \frac{\eta_a I_{ja} + \eta_p I_{jp} + I_{js} + \eta_h I_{jh}}{N_j}.$$

Integrating both sides, we get

$$\ln \frac{S_i(0)}{S_i(\infty)} = \Psi_{iS} a_i \beta \sum_{j=1}^{2} \frac{c_{ij} \Psi_{jI}}{N_j} \int_0^{\infty} \eta_a I_{ja} + \eta_p I_{jp} + I_{js} + \eta_h I_{jh} \mathrm{d}t, \qquad (B\,1)$$

where

$$\int_0^{\infty} I_{ia} \mathrm{d}u = \tau^A p^A (Z_i + E_{i0}), \quad \int_0^{\infty} I_{ip} \mathrm{d}u = \tau^P p^P (Z_i + E_{i0}) \Bigg\}$$

and

$$\int_0^{\infty} I_{is} \mathrm{d}u = \tau_i^S p^P (Z_i + E_{i0}), \quad \int_0^{\infty} I_{ih} \mathrm{d}u = \tau^H p^P p_i^H (Z_i + E_{i0}). \qquad (B\,2)$$

From (B 1) and (B 2), we obtain

$$\ln \frac{S_i(0)}{S_i(\infty)} = \Psi_{iS} a_i \beta \sum_{j=1}^{2} \frac{c_{ij} \Psi_{jI}}{N_j} (\eta_a \tau^A p^A + \eta_p \tau^P p^P + \tau_j^S p^P + \tau^H p^P p_j^H)(Z_j + E_{j0}),$$

which can be simplified to

$$\ln \frac{S_i(0)}{S_i(\infty)} = a_i \beta \sum_{j=1}^{2} A_{ij} \frac{Z_j + E_{j0}}{N_j}.$$

From the above equation, we get the following final size relation:

$$Z_i = S_i(0) \left( 1 - \exp \left[ -\sum_{j=1}^{2} a_i \beta A_{ij} \frac{Z_j + E_{j0}}{N_j} \right] \right), \quad i = 1, 2. \qquad (B\,3)$$

# Appendix C. Effect of allocation on the relationship between $\mathcal{F}$ and $\mathcal{R}_C$

Figure 8 shows the top right of figure 4 enlarged, with the addition of assuming 5% supplemental masks available to the total population. The two lines represent the range of possible allocations of masks and results for different quantities of supplemental masks. The two lines intersect, illustrating that for 5% and 10% availability, there exist an allocation for each scenario such that the resulting $\mathcal{R}_C$ and final size $\mathcal{F}$ are the same. Furthermore, the poor allocation of masks with 10% availability is worse than the proper allocation with 5% availability.

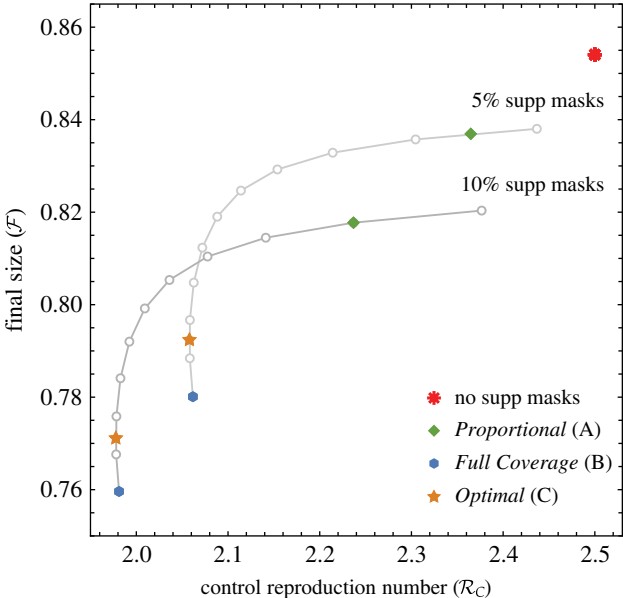

**Figure 8.** The effect of mask supply on the relationship between final size and control reproduction number. We assume 5% HCW and $\beta = 0.0332$.

# Appendix D. Sensitivity analysis

The sensitivity analyses provide information on how variation in parameters affects model results. The sensitivity analyses also show the most influential parameters on the reproduction numbers for our given set of parameter ranges, means and distribution. The parameter values and ranges used in the sensitivity analysis were estimated or found in the literature. We chose the probability density functions for each parameter based on the biology of COVID-19 and assigned a triangular distribution to all parameters except $p$, $c_1$ and $c_2$, which were assigned a uniform distributions. All parameters and assumptions used in the sensitivity analysis are in table 3, and the most important results are shown in figure 9.

# Appendix E. Other definitions

See table 4.

**Table 4.** Definitions and notation used.

| symbol | definition |
|---|---|
| $\lambda_i$ | force of infection for group $i$ |
| $\varsigma_{ij}$ | proportion of the $i$th sub-group's contacts that is with members of the $j$th |
| $N$ | total population size |
| $N_i$ | population size in group $i$ |
| $m$ | proportion of supplemental masks available to the total population |
| $q$ | proportion of total population that are healthcare workers |
| $z_i$ | proportion of masks available to members of group $i$ |
| $\tilde{z}_i$ | allocation of masks under Strategy **B** in group $i$ |
| $\hat{z}_i$ | allocation of masks under Strategy **C** in group $i$ |
| $\mathcal{F}$ | final epidemic size |
| $\mathcal{F}_i$ | fraction infected in group $i$ |
| $\mathcal{M}_i$ | total number of deaths in group $i$ |

*(Continued.)*

**Table 4.** (Continued.)

| symbol | definition |
| --- | --- |
| $Z_i$ | total number of people infected in group $i$ |
| $\mathcal{R}_C$ | control reproduction number |
| $\mathcal{R}_0$ | basic reproduction number |
| $\Delta^{\mathcal{R}}_{(z_1, z_2)}$ | efficacy of the Strategy **C** relative to the allocation $(z_1, z_2)$ with respect to reducing $\mathcal{R}_C$ |
| $\Delta^{\mathcal{F}}_{(z_1, z_2)}$ | efficacy of the Strategy **C** relative to the allocation $(z_1, z_2)$ with respect to reducing $\mathcal{F}$ |

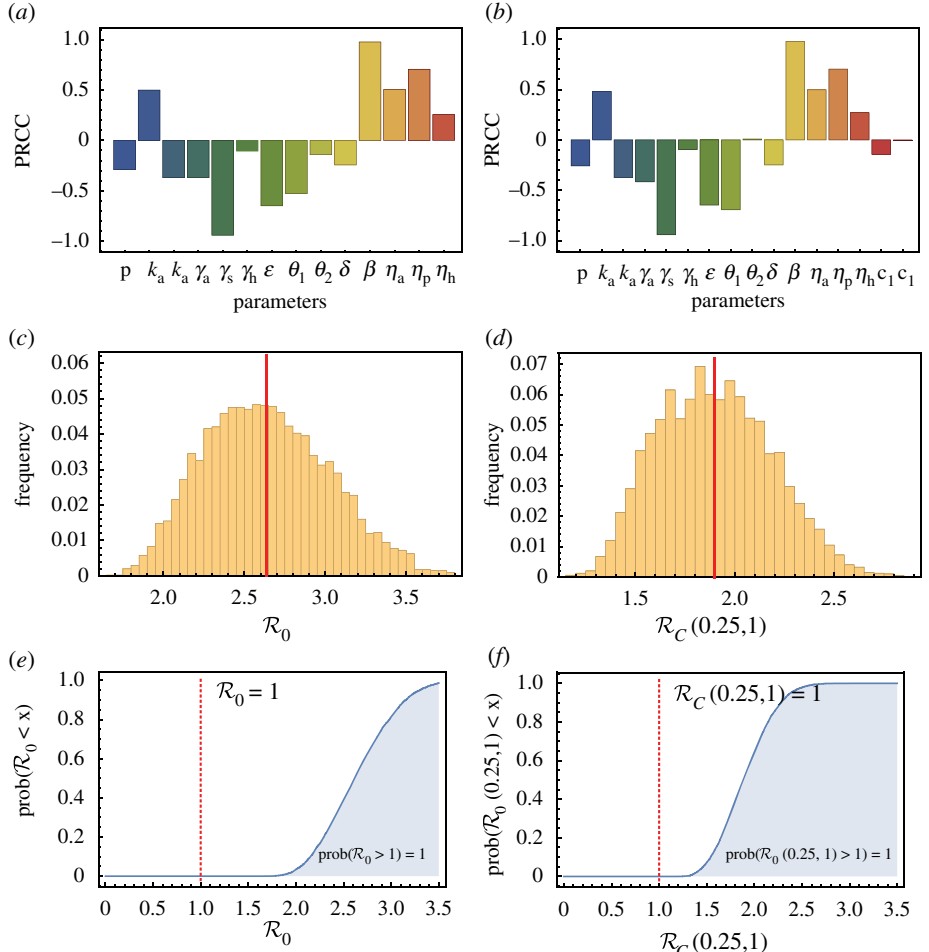

**Figure 9.** Sensitivity analyses of (a) $\mathcal{R}_0$ and (b) $\mathcal{R}_C(0.25, 1)$ with respect to model parameters. The plots show the distributions of (c) $\mathcal{R}_0$ and (d) $\mathcal{R}_C(0.25, 1)$, and the empirical cumulative distribution function of (e) $\mathcal{R}_0$ and (f) $\mathcal{R}_C(0.25, 1)$. Parameter values and ranges are listed in table 3.

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
