## [Peer Review File · Royal Society Open Science]

Review History

RSOS-210823.R0 (Original submission)

Review form: Reviewer 1 (Sanyi Tang)

Is the manuscript scientifically sound in its present form?

Yes

Are the interpretations and conclusions justified by the results?

Yes

Is the language acceptable?

Yes

Do you have any ethical concerns with this paper?

No

Have you any concerns about statistical analyses in this paper?

No

Recommendation?

Accept as is

Comments to the Author(s)

The submitted paper was well written and some interesting results concerning the COVID-19 control have been provided, I do suggest that this paper could be accepted for publication RSOS.

Review form: Reviewer 2**Is the manuscript scientifically sound in its present form?**

Yes

Are the interpretations and conclusions justified by the results?

Yes

Is the language acceptable?

Yes

Do you have any ethical concerns with this paper?

No

Have you any concerns about statistical analyses in this paper?

No

Recommendation?

Accept with minor revision (please list in comments)

Comments to the Author(s)

Comment on Manuscript_RSOS-210823

In this manuscript, the authors developed an ODEs system consisting of two groups (HCW and members of the community) and investigated the optimal allocation of potentially scarce medical resources between these two groups to control the pandemic. The manuscript overall is well-written, and the methods and findings are illustrated with clear details. However, some concerns can be addressed to increase clarification:

1. My main concern is about the model setting. Many studies have studied the transmission of diseases in a hospital setting (e.g., <https://doi.org/10.1098/rsos.201895>; <https://doi.org/10.1371/journal.pone.0030170>; doi:10.3934/mbe.2019181; doi:10.3934/mbe.2015.12.761<https://doi.org/10.1016/j.mbs.2019.01.013>) that the three main causes for HCW infections are due to 1. HCW-to-HCW contacts; 2 HCW-to-Patient contacts; 3, environmental contamination. For the specialty of COVID-19, I am ok to consider HCW-to-friends/family/neighbors/other members of the community contact. But please clarify why you ignore patients (as another important group) if you try to control the infections for HCWs, since in my view if you ignore patients, you will greatly decrease the risk of HCWs. Is it a better idea to consider HCWs, patients, and other communities? Please clarify.
2. The authors used both the reproduction number and final size as measurements for control of the disease for different allocation strategies. As we know, the population of HCWs is relatively small compared to the general community). The final size $F = F_1 + F_2 = Z_1/N$ (the fraction of HCWs) $+ Z_2/N$ (the fraction of community) is not fair to use since even though all HCWs are infected, F value can't be reflected dramatically. I would suggest using both F_1 and F_2 respectively to indicate the control of the disease since it is a big deal if the hospital's system is ruined by pandemics (e.g., most HCWs get infections). So it goes back to my #1 concern, please clarify the model set.

3. For table 1, it is best to have another column to indicate the source/reference of your parameter values even though in the text you mentioned some reference there.
4. For figure 3, I think you showed the result for Optimal strategies B and C. Please change your figure legend (A with % HCW; B with % HCW) to B and C.

Decision letter (RSOS-210823.R0)

Dear Dr Feng

On behalf of the Editors, we are pleased to inform you that your Manuscript RSOS-210823 "Optimal allocation of resources to health care workers or the general populace: a modeling study" has been accepted for publication in Royal Society Open Science subject to minor revision in accordance with the referees' reports. Please find the referees' comments along with any feedback from the Editors below my signature.

Please submit your revised manuscript and required files (see below) no later than 7 days from today's (ie 25-Oct-2021) date. Note: the ScholarOne system will 'lock' if submission of the revision is attempted 7 or more days after the deadline. If you do not think you will be able to meet this deadline please contact the editorial office immediately.

on behalf of Dr Shigui Ruan (Associate Editor) and Glenn Webb (Subject Editor)
openscience@royalsociety.org

Associate Editor Comments to Author (Dr Shigui Ruan):

Comments to the Author:

Please revise your manuscript by addressing the comments made by the second reviewer.

Reviewer comments to Author:

Reviewer: 1

Comments to the Author(s)

The submitted paper was well written and some interesting results concerning the COVID-19 control have been provided, I do suggest that this paper could be accepted for publication RSOS.

Reviewer: 2

Comments to the Author(s)

Comment on Manuscript_RSOS-210823

In this manuscript, the authors developed an ODEs system consisting of two groups (HCW and members of the community) and investigated the optimal allocation of potentially scarce medical resources between these two groups to control the pandemic. The manuscript overall is well-written, and the methods and findings are illustrated with clear details. However, some concerns can be addressed to increase clarification:

1. My main concern is about the model setting. Many studies have studied the transmission of diseases in a hospital setting (e.g., <https://doi.org/10.1098/rsos.201895>; <https://doi.org/10.1371/journal.pone.0030170>; doi:10.3934/mbe.2019181; doi:10.3934/mbe.2015.12.761<https://doi.org/10.1016/j.mbs.2019.01.013>) that the three main causes for HCW infections are due to 1. HCW-to-HCW contacts; 2 HCW-to-Patient contacts; 3, environmental contamination. For the specialty of COVID-19, I am ok to consider HCW-to-friends/family/neighbors/other members of the community contact. But please clarify why you ignore patients (as another important group) if you try to control the infections for HCWs, since in my view if you ignore patients, you will greatly decrease the risk of HCWs. Is it a better idea to consider HCWs, patients, and other communities? Please clarify.

2. The authors used both the reproduction number and final size as measurements for control of the disease for different allocation strategies. As we know, the population of HCWs is relatively small compared to the general community). The final size $F=F_1+F_2=Z_1/N$ (the fraction of HCWs) $+Z_2/N$ (the fraction of community) is not fair to use since even though all HCWs are infected, F value can't be reflected dramatically. I would suggest using both F_1 and F_2 respectively to indicate the control of the disease since it is a big deal if the hospital's system is ruined by pandemics (e.g., most HCWs get infections). So it goes back to my #1 concern, please clarify the model set.

3. For table 1, it is best to have another column to indicate the source/reference of your parameter values even though in the text you mentioned some reference there.

4. For figure 3, I think you showed the result for Optimal strategies B and C. Please change your figure legend (A with % HCW; B with % HCW) to B and C.

===PREPARING YOUR MANUSCRIPT===

one version should clearly identify all the changes that have been made (for instance, in coloured highlight, in bold text, or tracked changes);
 a 'clean' version of the new manuscript that incorporates the changes made, but does not highlight them. This version will be used for typesetting.

===PREPARING YOUR REVISION IN SCHOLARONE===

-- Ensure that your data access statement meets the requirements at https://royalsociety.org/journals/authors/author-guidelines/#data.

You should ensure that you cite the dataset in your reference list. If you have deposited data etc in the Dryad repository, please only include the 'For publication' link at this stage. You should remove the 'For review' link.

-- If you are requesting an article processing charge waiver, you must select the relevant waiver option (if requesting a discretionary waiver, the form should have been uploaded, see 'File upload' above).

-- If you have uploaded any electronic supplementary (ESM) files, please ensure you follow the guidance at <https://royalsociety.org/journals/authors/author-guidelines/#supplementary-material> to include a suitable title and informative caption. An example of appropriate titling and captioning may be found at https://figshare.com/articles/Table_S2_from_Is_there_a_trade-off_between_peak_performance_and_performance_breadth_across_temperatures_for_aerobic_scope_in_teleost_fishes_/3843624.

Author's Response to Decision Letter for (RSOS-210823.R0)

See Appendix A.

Decision letter (RSOS-210823.R1)

Dear Dr Feng,

I am pleased to inform you that your manuscript entitled "Optimal allocation of resources to health care workers or the general populace: a modeling study" is now accepted for publication in Royal Society Open Science.

COVID-19 rapid publication process:

We are taking steps to expedite the publication of research relevant to the pandemic. If you wish, you can opt to have your paper published as soon as it is ready, rather than waiting for it to be published the scheduled Wednesday.

This means your paper will not be included in the weekly media round-up which the Society sends to journalists ahead of publication. However, it will still appear in the COVID-19 Publishing Collection which journalists will be directed to each week (<https://royalsocietypublishing.org/topic/special-collections/novel-coronavirus-outbreak>).

If you wish to have your paper considered for immediate publication, or to discuss further, please notify openscience_proofs@royalsociety.org and press@royalsociety.org when you respond to this email.

on behalf of Glenn Webb (Subject Editor)
openscience@royalsociety.org

Appendix A

Responses to Reviewer 2

We thank the reviewers for their comments and suggestions that helped us to improve the presentation of our work. Reviewer's comments are italicized.

Comments:

1. *My main concern is about the model setting. Many studies have studied the transmission of diseases in a hospital setting (e.g., <https://doi.org/10.1098/rsos.201895>; <https://doi.org/10.1371/journal.pone.0030170>; [doi:10.3934/mbe.2019181](https://doi.org/10.3934/mbe.2019181); [doi:10.3934/mbe.2015.12.761](https://doi.org/10.3934/mbe.2015.12.761) <https://doi.org/10.1016/j.mbs.2019.01.013>) that the three main causes for HCW infections are due to 1. HCW-to-HCW contacts; 2 HCW-to-Patient contacts; 3, environmental contamination. For the specialty of COVID-19, I am ok to consider HCW-to-friends/family/neighbors/other members of the community contact. But please clarify why you ignore patients (as another important group) if you try to control the infections for HCWs, since in my view if you ignore patients, you will greatly decrease the risk of HCWs. Is it a better idea to consider HCWs, patients, and other communities? Please clarify.*

Response: We added references to other modeling studies concerning HCWs. Our focus is on identifying the optimal allocation under limited supply of masks between HCWs and the general community. But our model includes four infectious states, one of which is symptomatic people who are hospitalized (i.e., patients), so there are contacts between HCWs and patients.

2. *The authors used both the reproduction number and final size as measurements for control of the disease for different allocation strategies. As we know, the population of HCWs is relatively small compared to the general community). The final size $F=F1+F2=Z1/N$ (the fraction of HCWs) + $Z2/N$ (the fraction of community) is not fair to use since even though all HCWs are infected, F value can't be reflected dramatically. I would suggest using both $F1$ and $F2$ respectively to indicate the control of the disease since it is a big deal if the hospital's system is ruined by pandemics (e.g., most HCWs get infections). So it goes back to my #1 concern, please clarify the model set.*

Response: In this study, we considered the proportions of infected people as a measure to compare various strategies. But the reviewer's point is well taken. In future studies, we might also consider HCWs as a potentially scarce resource. We discuss our choice and possible future work in the last paragraph.

3. *For table 1, it is best to have another column to indicate the source/reference of your parameter values even though in the text you mentioned some reference there.*

Response: Done.

4. *For figure 3, I think you showed the result for Optimal strategies B and C. Please change your figure legend (A with % HCW; B with % HCW) to B and C.*

Response: The legend now reads: Strategy C is compared with Strategies A and B. The figure is correct.